# Anti-Cancer Immune Reaction and Lymph Node Macrophage; A Review from Human and Animal Studies

**Yoshihiro Komohara** [1,*,†]**, Toshiki Anami** [1,2,†]**, Kenichi Asano** [3]**, Yukio Fujiwara** [1]**, Junji Yatsuda** [2] **and Tomomi Kamba** [2]

1    Department of Cell Pathology, Graduate School of Medical Sciences, Kumamoto University, Kumamoto 860-0862, Japan; toshiki_anami@yahoo.co.jp (T.A.); fuji-y@kumamoto-u.ac.jp (Y.F.)
2    Department of Urology, Graduate School of Medical Sciences, Kumamoto University, Kumamoto 860-0862, Japan; jun_yatsuda@hotmail.co.jp (J.Y.); kamba@kumamoto-u.ac.jp (T.K.)
3    Laboratory of Immune Regulation, School of Life Sciences, Tokyo University of Pharmacy and Life Sciences, Tokyo 192-0392, Japan; asanok@toyaku.ac.jp
*    Correspondence: ycomo@kumamoto-u.ac.jp
†    First two authors equally contributed to this work.

**Abstract:** Lymph nodes are secondary lymphoid organs that appear as bean-like nodules usually <1 cm in size, and they are localized throughout the body. Many antigen-presenting cells such as dendritic cells and macrophages reside in lymph nodes, where they mediate host defense responses against pathogens such as viruses and bacteria. In cancers, antigen-presenting cells induce cytotoxic T lymphocytes (CTLs) to react to cancer cell-derived antigens. Macrophages located in the lymph node sinus are of particular interest in relation to anti-cancer immune responses because many studies using both human specimens and animal models have suggested that lymph node macrophages expressing CD169 play a key role in activating anti-cancer CTLs. The regulation of lymph node macrophages therefore represents a potentially promising novel approach in anti-cancer therapy.

**Keywords:** macrophage; lymph node; CD169; PD-L1

## 1. The Critical Role of Lymph Nodes in Anti-Cancer Immunotherapy

Cancer cells are characterized by the accumulation of a variable number of genetic alterations that result in the production of neoantigens. Cancer specific antigens such as cancer-testis antigens, oncofetal antigens, aberrantly expressed proteins, and viral antigens are also targets of cytotoxic T cells. $CD8^+$ T cells recognize cancer cells via binding between the T-cell receptor and major histocompatibility complex class I/peptide complex [1]. Chen and Mellman (2013) suggested that the immune system is triggered to eliminate cancer cells via stimulation of the cancer-immunity cycle [2]. Immune checkpoint blockade therapy targeting cytotoxic T-lymphocyte (CTL)-associated antigen 4 (CTLA-4) or programmed death 1 (PD-1)/programmed death ligand 1 (PD-L1) has become a promising anti-cancer immunotherapy approach [3]. Anti-PD-1 and anti-CTLA-4 therapy are reportedly effective for patients with several types of solid tumors, such as melanoma, non-small-cell lung cancer, renal cell carcinoma, urothelial carcinoma, head and neck squamous cell carcinoma (SCC), esophageal SCC, gastric adenocarcinoma, triple-negative breast carcinoma, and microsatellite instability-high tumors [4].

CTLA-4 is expressed on T-lymphocytes and competitively inhibits the binding of CD28 to costimulatory molecules such as CD80 and CD86. PD-1 ligands are expressed on both cancer cells and immune cells. Among immune cells, antigen-presenting cells such as macrophages and dendritic cells (DCs) express high levels of PD-1 ligands [5]. Myeloid cells express PD-1 ligands in both the tumor microenvironment and lymph nodes [6]. Fransen et al. demonstrated that $CD11b^+$ myeloid cells residing in lymph nodes express significantly higher levels of PD-L1 in tumor-bearing mice; lymph node resection in these mice abrogated the anti-tumor effect of anti–PD-1 therapy [7]. Fransen et al. also found that

the anti-tumor effect of anti-PD-1 therapy was suppressed by the S1P receptor inhibitor FTY720, which restricts T cells in lymphoid organs. Zhao et al. reported that lymph node resection in the early stage abrogated anti-tumor immune responses, however, immune responses were not observed by lymph node resection in the advanced stage [8]. They additionally showed that anti-tumor immune cells were restricted to tumor-draining lymph nodes in the early stage and spread to the spleen in the advanced stage, which indicated that surgical resection of regional lymph nodes in patients with advanced tumors might not affect anti-tumor immunity in patients. Dammeijer et al. detected expression of PD-L1 in DCs and macrophages in tumor-draining lymph nodes; blocking PD-L1 on DCs (but not macrophages) induced an effective anti-tumor immune response [9]. These authors also found greater interaction between PD-1 and PD-L1 in the lymph nodes than tumors, which correlated with shorter relapse-free survival. These findings suggest that tumor-draining lymph nodes play a critical role in the anti-tumor immune responses induced by anti-PD1/PD-L1 therapy in the early stage of diseases.

## 2. Function of Lymph Node Macrophages in Mice

Interest in the role of lymph node macrophages in the initiation of immune responses is increasing [10]. Research in this area has shown that lymph node sinus macrophages (SMs) express sialoadhesin (CD169), a 185-kDa type I lectin involved in phagocytosis of pathogens and cell–cell contact with lymphocytes via binding to CD43 (sialophorin). SMs are divided into two subtypes in mice: subcupsular SMs (SCSMs) and medullary SMs (MSM), which are characterized as CD11b$^+$ CD169$^+$ F4/80$^-$ and CD11b$^+$ CD169$^+$ F4/80$^+$ cells, respectively (Figure 1A) [11,12]. However, macrophages located in the medullary cord in both humans and rodents are CD169$^-$. CD169 expression is specifically restricted to macrophages, particularly resident macrophages in the spleen, liver, bone marrow, and intestines. CD169 is thought to play a role in the uptake of sialylated antigens and is therefore considered a potentially useful target for antigen delivery in vaccine development in mice [13,14]. Targeting antigen delivery to CD169-expressing cells was also shown to enhance immune responses in pigs [15]. These animal model data thus indicate that SCSMs/MSMs can take up and present tumor antigens to lymphocytes, thus inducing effective immune responses.

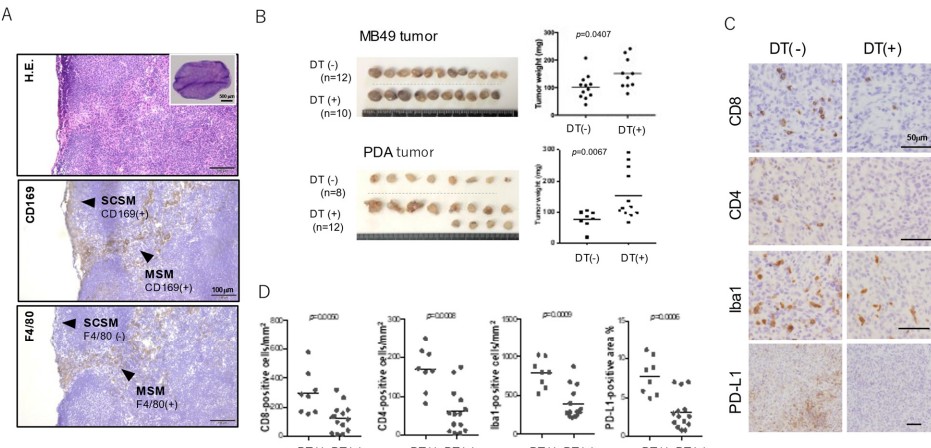

**Figure 1.** In vivo murine tumor model in CD169-DTR mice. (**A**) Representative figure of hematoxylin and eosin staining and immunohistochemical staining of CD169 (brown colored) of murine lymph nodes. (**B**) Following subcutaneous injection of diphtheria toxin (DT) on day 0, mice were inoculated subcutaneously with $5 \times 10^5$ tumor cells (day 1). Subcutaneous tumor nodules were dissected after 10 days (day 11). (**C**) Tumor samples were fixed and embedded in paraffin. Sections were stained with anti-CD8, anti-CD4, anti-Iba1 (marker for pan-macrophage), and anti–PD-L1 antibodies. (**D**) Cell count and signal value data were evaluated using ImageJ software. Differences between the two groups were evaluated using the Mann–Whitney *U*-test.

## 3. The Function of Lymph Node Macrophages in the Anti-Tumor Immune Response in Mice

Tumor tissues, particularly malignant tumors with a high proliferative capacity, contain numerous apoptotic or necrotic dead cells. Tumor cell debris or tumor cell antigens that migrate to the draining lymph nodes are taken up by antigen-presenting cells such as DCs, which facilitate recognition of tumor antigens by naïve CD8[+] CTLs and induce tumor cell–specific CTLs [16]. Das Mohapatra et al. reported that apoptotic tumor cells are taken up through phagocytosis by DCs and SM, whereas live tumor cells are taken up via trogocytosis, primarily by DCs [17]. The critical role of SMs in anti-cancer immune responses is evidenced by in vivo studies using animal models. Subcutaneous injection of dead cancer cells is known to induce tumor antigen-specific CTLs [18]. To explore the function of CD169-expressing macrophages in anticancer immune responses, Asano et al. investigated the effect of depletion of macrophages in diphtheria toxin receptor (DTR) transgenic mice in which CD169[+] macrophages were specifically depleted by diphtheria toxin abrogated anti-cancer immune responses in mice vaccinated with dead cancer cells. [19]. However, there was no difference in the CTL response to tumor cells between wild-type and CD169-deficient mice, indicating that CD169 does not play a critical role in the anti-tumor immune response.

CD169 expression is restricted to marginal zone metallophilic macrophages in the spleen [20]. Antigens circulating in the blood flow are preferentially captured by spleen antigen presenting cells including CD169[+] macrophages which transfer the antigens to B cells and DCs [21,22]. Benhard et al. demonstrated that both DCs and CD169[+] macrophages induce CTL responses in the spleen. They found that DCs induce CTLs that react to strongly binding epitopes, whereas macrophages induce CTLs that react to a broader range of epitopes [23]. Muraoka et al. demonstrated that CHP nanogel-conjugated tumor antigens are specifically engulfed by MSMs, induce antigen-specific CTLs, and suppress tumor development [24]. These findings indicate that MSMs, in addition to SCSMs and DCs, also function in antigen presentation.

We attempted to confirm the suppression of tumor development in another murine tumor model using MB49 (murine bladder cancer) and PDA (pancreatic ductal adenocarcinoma) cell lines and CD169-DTR mice. Consistent with the previous studies described above, the growth of MB49 and PDA subcutaneous tumors was significantly promoted by macrophage-depletion via diphtheria toxin (Figure 1B). Immunohistochemical analysis using paraffin-embedded tumor samples showed a significant increase in infiltrating lymphocytes in tumor tissues in SM-depleted mice (Figure 1C,D). Macrophage infiltration and PD-L1 expression were significantly lower in SM-depleted mice as compared with control mice, suggesting that the tumor microenvironment in SM-depleted mice has characteristics of non-inflamed tumors.

## 4. The Function of SMs in Protumor Function of B-Lymphocytes in Mice

Saunderson et al. first reported that CD169[+] macrophages in the spleen and lymph node captured exosomes via binding to CD169. They also reported that microvesicles were not retained in the subcapsular sinus of CD169-deficient mice but penetrated deeper into the lymph node paracortex [25]. Muhsin-Sharafaldine suggested that CD169 mediates anti-tumor immune responses via the uptake of tumor antigen-containing microvesicles; however, the anti-tumor immune response was not affected in CD169-deficient animals [26]. Pucci et al. reported an increase in B cells in draining lymph nodes following SM depletion and that pro-tumor immunoglobulin was involved in tumor growth [27]. They also found that the capture of tumor-derived exosomes by SMs inhibited the proliferation of pro-tumor B cells in the lymph nodes; an immunohistochemical analysis using human samples revealed the presence of melanoma-derived antigens in melanoma-free draining lymph nodes. Similar effects of SMs on B-cell proliferation in murine 4T1 and MMTV-PyMT mammary carcinoma models were reported by Tacconi et al. [28]. These authors demonstrated lung metastasis of cancer cells in SM-depleted mice, but depletion of B cells using an anti-CD20 antibody significantly suppressed lung metastasis. RNA sequencing of SMs in tumor-draining lymph nodes revealed the up-regulation of several genes potentially associated with B-cell activation.

## 5. Potential Origin of SMs in Mice

The M1/M2 classification based on the condition of macrophage activation was introduced in the late 1990s [29,30]. A Th1-like immune environment is generally believed to induce M1(-like) macrophage polarization, whereas a Th2-like immune environment induces M2(-like) macrophage polarization. SMs express both M1- and M2-related genes, making these cells unsuitable for M1/M2 classification [31]. In addition to M1/M2 classification, macrophages can be categorized into two types depending on origin: monocyte-derived macrophages and tissue-resident macrophages derived from hematopoietic precursors in the embryonic yolk sac or fetal liver at birth [32–34]. Kupffer cells in the liver, alveolar macrophages in the lung, microglia in the brain parenchyma, as well as macrophages in the intestines, pancreas, and abdominal cavity, are classified as tissue-resident macrophages; however, the origin of SMs remains unclear [35]. Tacconi et al. reported that both SCSMs and MSMs exhibit proliferative activity and that inhibition of CSF1R abrogated SM proliferation in tumor-draining lymph nodes. Based on the results of parabiosis tests, Pucci et al. suggested that SMs are tissue-resident macrophages. Investigations of cell–cell interaction between B lymphocytes and SMs indicated that SCSM proliferation is dependent on lymphotoxin (LT)-$\alpha 1/\beta 2$ secreted from B lymphocytes [36]. Selective transgenic overexpression of LT$\alpha 1/\beta 2$ in B cells led to an increase in the total number of SCSMs without affecting the number of B cells [37]. These observations suggest that both SCSMs and MSMs have proliferative and self-renewal activity similar to tissue-resident macrophages originating from embryonic precursors. However, what causes the difference in F4/80 antigen expression between SCSMs and MSMs remains to be determined.

## 6. The Importance of SM in Human Solid Tumors

Routine pathological analyses afford many opportunities for clinicians to observe the lymph nodes of cancer patients. The presence of metastasis to the lymph nodes is an important factor in determining disease stage, and the pathologist is responsible for making that decision. We examined CD169 expression in human lymph nodes immunohistochemically using paraffin-embedded sections. SMs were also positive for other macrophage markers such as CD163 and CD204, and Ki67-positive SMs were also observed (Figure 2). The DC-related marker fascin was also expressed on SMs [38]. First, when we analyzed samples from cases of colorectal carcinoma, and interestingly, the rate of CD169 positivity of the SMs varied greatly among individual cancer-bearing patients and non-cancer controls [39]. This observation was inconsistent with CD169 expression in mouse lymph nodes. There were also no clear differences between SCSMs and MSMs, in contrast to mouse lymph nodes. Statistical analyses of postoperative survival and clinicopathological factors between patients with high and low numbers of CD169-positive SMs in lymph nodes were also performed. In cases with colorectal carcinoma, high CD169 expression in SMs had significantly longer overall survival, smaller tumor size, and less lymph node metastasis [39]. Multivariate analysis identified CD169 positivity rate in SMs as an independent factor in determining overall survival in colorectal carcinoma patients. The same analysis was performed for malignant melanoma, bladder cancer, endometrial carcinoma, gastric cancer, and esophageal cancer, and in cases with high CD169 expression, a significant prolongation of overall survival and cancer-specific survival was observed [40–44]. Multivariate analysis also indicated that a high number of CD169[+] SMs is an independent prognostic factor in malignant melanoma, gastric cancer, and bladder cancer. Other researchers have reported similar results for prostate cancer and breast cancer [45,46]. Infiltrating CD8-positive T cells in tumor tissues, particularly in tumor nests, play a central role in the anti-tumor immune response in cancer patients. We hypothesized that CD169[+] SMs activate CD8-positive T cells to promote anti-tumor immunity and therefore analyzed CD169[+] SMs and the infiltration of CD8-positive T cells into tumor tissues. The number of CD8-positive T cells in tumor tissues was significantly higher in colorectal carcinoma, malignant melanoma, gastric cancer, breast cancer, endometrial carcinoma, and bladder cancer, which exhibited a high CD169 positivity rate in SMs (Table 1). In endometrial carcinoma, the number of CD169[+] SMs was also correlated with infiltration of NK cells into the tumor [41].

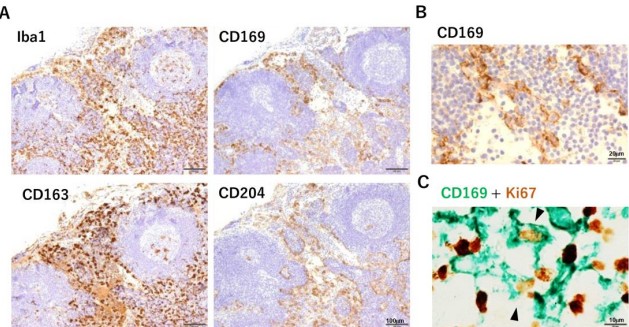

**Figure 2.** Immunohistochemical analysis of human lymph nodes. (**A**) SMs were positive for CD169, CD163, and CD204. Lymphatic endothelial cells were also positive for CD204. Scale bar, 100 μm. (**B**) High-power field examination of the lymph sinus. The cell membrane of macrophages was positive for CD169. Scale bar, 20 μm. (**C**) Double immunohistochemical staining of CD169 (green) and Ki67 (brown) in the lymph node sinus. Some double-positive cells were observed. Scale bar, 10 μm.

The results of the present study demonstrate a significant correlation between CD169[+] SMs in anti-tumor immune responses and a better clinical course. Other groups have also suggested that CD169[+] SMs play a critical role in anti-tumor immune responses (Table 1). Gunnarsdottir et al. reported that the co-localization of CD169[+] SMs and cancer cells in lymph node metastatic lesions was linked to improved recurrence-free survival in patients with breast cancer [46]. Those authors also examined PD-L1 expression in SMs; however, they found no significant association between PD-L1 expression on SMs and clinical course [46]. Using a rat prostate cancer model, Strömvall et al. identified several genes, including the gene encoding CD169, which are up-regulated in the pre-metastatic niche in tumor-draining lymph nodes [49]. In a subsequent study using the rat model, they found fewer CD169[+] SMs in pre-metastatic tumor-draining lymph nodes, and a reduction in the number of CD169[+] SMs was found to be closely associated with shortened relapse-free survival in prostate cancer patients [45]. Topf et al. reported that the metastatic spread of head and neck carcinoma to regional lymph nodes was associated with fewer CD169[+] SMs in draining lymph nodes [48]. This reduction in the number of SMs was significant in cancer cases without human papilloma virus infection. Thus, accumulating evidence suggests that CD169[+] SMs play a critical role in determining the clinical course of various types of cancer, but details regarding the relationship between SMs and cancer in humans remain to be determined.

**Table 1.** Clinical significance of CD169[+] sinus macrophages (SMs) in solid tumors.

| Tumor Type | Link to Prognosis | Link to TIL | Comments | Reference |
|---|---|---|---|---|
| Colorectal cancer | better OS | yes | High density of CD169[+] SMs were observed in cases with T1/2 stage or without lymph node metastasis. CD169 expression in human monocyte-derived macrophages was increased by type-I interferons (IFNs). | [39] |
| Melanoma | better OS | yes | IFN-alpha secreted from plasmacytoid dendritic cells was suggested to link to CD169 expression. | [40] |
| Endometrial cancer | better OS | yes | High density of CD169[+] SMs were correlated with high density of natural killer cells more significantly than that of TILs. | [41] |
| Bladder cancer | better OS | yes | High density of CD169[+] SMs were observed in cases with T1/2 stage. | [42] |
| Esophageal cancer | better OS | yes | The density of CD169[+] SMs were higher in female than male. Significant correlation between CD169[+] SMs and TILs were seen in cases with neoadjuvant therapy. CD169[+] SMs partially expressed IDO1. | [44] |
| Breast cancer | not significant | yes | High density of CD169[+] SMs was correlated with high density of TILs in cases with high Ki67 index. | [47] |
| Breast cancer | better PFS | not done | High density of CD169[+] SMs were seen in cases with low tumor size, and correlated with PD-L1 expression both in primary tumor and metastatic tumor. Co-expression of CD169 and PD-L1 was seen in cases of younger age. | [46] |

**Table 1.** *Cont.*

| Tumor Type | Link to Prognosis | Link to TIL | Comments | Reference |
|---|---|---|---|---|
| Gastric cancer | better OS and PFS | yes | High density of CD169+ SMs was associated with better OS in cases with advanced cancer or without metastasis and correlated to better PFS in cases with diffuse type or high tumor-stroma ratio. | [43] |
| Prostate cancer | better OS | not done | The significance of CD169+ SMs was suggested by rat prostate cancer metastatic model. | [45] |
| Head and neck cancer | not done | not done | High density of CD169+ SMs were observed in lymph node without metastasis. | [48] |

OS: cancer specific overall survival, PFS; progression free survival; TIL; tumor infiltrating lymphocytes, IDO; indoleamine-2,3-dioxygenase.

## 7. Conclusions

Evidence indicating that CD169+ SMs play a significant role in anti-cancer immune responses is increasing, and many studies examining human samples and mouse models have been published (Figure 3). Targeted delivery of anti-tumor vaccines to SMs is also considered an effective anti-tumor vaccine therapy approach. The mechanism underlying the observed individual variation in the number of CD169+ SMs in lymph nodes remains unknown. Analyzing CD169 in regional lymph nodes could both help predict the clinical prognosis in patients with several types of solid tumors as well as enable the prediction of anti-cancer immune responses. With regard to the observed individual variability in the number of CD169+ SMs in human samples, we suggest that this variability derives from a different population of tissue-resident macrophages originating from embryonic precursors.

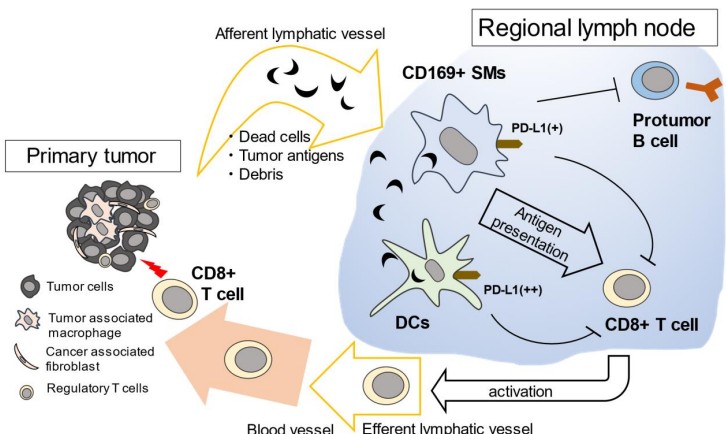

**Figure 3.** Proposed model of the function of CD169+ SMs in anti-tumor immunity. Dead tumor cells and debris are delivered to draining lymph nodes via lymphatic vessels. CD169+ LySMs in the regional lymph nodes capture dead tumor cells and present the tumor antigens to CD8+ T cells. SMs are also thought to suppress the proliferation of pro-tumor B cells. DCs capture tumor antigens passed through by SMs and activate tumor-specific CD8+ T cells, subsequently activated CD8+ T cells travel to the tumor microenvironment. Since both SMs and DCs express PD-L1, it is thought that anti-tumor immune reactions in the lymph nodes could be activated by anti-PD1/PD-L1 therapy.

**Author Contributions:** Data curation, Y.K. and T.A.; Funding acquisition, Y.K.; Investigation, Y.K. and T.A.; Methodology, K.A. and Y.F.; Resources, K.A. and Y.F.; Supervision, J.Y. and T.K.; Writing—Original draft, Y.K. and T.A.; Writing—Review and editing, J.Y. and T.K. All authors have read and agreed to the published version of the manuscript.

**Funding:** This work was supported by grants from the Ministry of Education, Culture, Sports, Science and Technology of Japan (Nos20H03459 to Y.K.).

**Institutional Review Board Statement:** All animal procedures were planned according to the AR-RIVE guidelines and approved by the Animal Research Committee at Kumamoto University (#A2020-094). Paraffin-embedded lymph node samples were prepared from specimens obtained from patients diagnosed with colon cancer and surgically resected at Izumi General Hospital (Izumi, Kagoshima,

Japan). Written informed consent was obtained from all patients, and the study design was approved by the review board (#57).

**Informed Consent Statement:** Informed consent was obtained from all subjects involved in the study.

**Conflicts of Interest:** None of the authors have any conflicts of interest in association with this manuscript.

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
