# Peer review of "Anti-Cancer Immune Reaction and Lymph Node Macrophage; A Review from Human and Animal Studies"

_2673-5601, doi:10.3390/immuno1030014_

Round 1

Reviewer 1 Report

The review article ‘Anti-cancer immune reaction and lymph node macrophage; a review from human and animal studies’ by Komohara et al. summarized the tumor-draining lymph nodes and CD169+ macrophage functions in anti-tumor immunity.

Major comments:

The authors missed important information to introduce tumor-draining lymph nodes in cancer immunity and immunotherapy response. The authors only cited the positive effects of tumor-draining lymph nodes in cancer immunotherapy. However, a recent study (Chemotherapy but not the tumor draining lymph nodes determine the immunotherapy response in secondary tumors, Link: https://pubmed.ncbi.nlm.nih.gov/32344378/) has shown that the functions of tumor-draining lymph nodes on cancer immunotherapy are disease-stage dependent. Please include these new findings as a small paragraph in the introduction to have a complete review of the tumor-draining lymph nodes' function. Remember that most tumor radical surgeries will resect tumor-draining lymph nodes (a lot of them are tumor-free after pathological exam), will this damage the efficacy of immunotherapy as an adjuvant treatment? If the tumor-draining lymph nodes are so critical for cancer immunity, should the surgeons stop resecting them? Please read and discuss the recommend article and think about different angles of tumor-draining lymph nodes in cancer immunity and immunotherapy.

The second major concern is that CD169+ macrophages are not specific to tumor-draining lymph nodes. It is hard to conclude that the effects of CD169+ macrophages are associated with the tumor-draining lymph nodes. How can you differentiate the functions of CD169+ macrophages in the spleen and tumor-draining lymph nodes? This point has to been discussed.

Author Response

We sincerely appreciate for kind review and helpful comments. We added the sentences as reviewer’s comments/advises. 

Comment 1:  The authors missed important information to introduce tumor-draining lymph nodes in cancer immunity and immunotherapy response. The authors only cited the positive effects of tumor-draining lymph nodes in cancer immunotherapy. However, a recent study (Chemotherapy but not the tumor draining lymph nodes determine the immunotherapy response in secondary tumors, Link: https://pubmed.ncbi.nlm.nih.gov/32344378/) has shown that the functions of tumor-draining lymph nodes on cancer immunotherapy are disease-stage dependent. Please include these new findings as a small paragraph in the introduction to have a complete review of the tumor-draining lymph nodes' function. Remember that most tumor radical surgeries will resect tumor-draining lymph nodes (a lot of them are tumor-free after pathological exam), will this damage the efficacy of immunotherapy as an adjuvant treatment? If the tumor-draining lymph nodes are so critical for cancer immunity, should the surgeons stop resecting them? Please read and discuss the recommend article and think about different angles of tumor-draining lymph nodes in cancer immunity and immunotherapy.

>> We added following sentences in the text as cited the reference.

“Zhao et al. reported that lymph node resection in early stage abrogated anti-tumor immune responses, however, immune responses was not observed by lymph node resection in advanced stage [8]. They additionally showed anti-tumor immune cells restricted in tumor-draining lymph node in early stage spread in spleen in advanced stage, and these indicated surgical resection of regional lymph nodes in patients with advanced tumor stages might not affect anti-tumor immunity in patients.”

Comment 2: The second major concern is that CD169+ macrophages are not specific to tumor-draining lymph nodes. It is hard to conclude that the effects of CD169+ macrophages are associated with the tumor-draining lymph nodes. How can you differentiate the functions of CD169+ macrophages in the spleen and tumor-draining lymph nodes? This point has to been discussed.

>> We added a sentence in the texts as follows

CD169 expression is restricted to marginal zone metallophilic macrophages in the spleen [20]. “ Antigens circulating in the blood flow preferentially captured by spleen antigen presenting cells including CD169+ macrophages which transfer the antigens to B cells and DCs [21,22].” Benhard et al. demonstrated that both DCs and CD169+ macrophages induce CTL responses in the spleen. They found that DCs induce CTLs that react to strongly binding epitopes, whereas macrophages induce CTLs that react to a broader range of epitopes [23]. Muraoka et al. demonstrated that CHP nanogel–conjugated tumor antigens are specifically engulfed by MSMs, induce antigen-specific CTLs, and suppress tumor development [24]. These findings indicate that MSMs, in addition to SCSMs and DCs, also function in antigen presentation.

Reviewer 2 Report

This review is well-written and is an appropriate review of the field.

Minor comments:

First sentence: Please revise, as there are more types of cancer antigens than those that derive from neoantigens and cancer testis antigens.

Section 2 could use more content reviewing macrophage subsets and function in lymph nodes. The section as is seems to primarily focus on CD169 expression. What are the different subsets, where do they localize, and how do they differ in function?

In section 6, it is a little unclear if the authors are referring to high CD169 expression on SMs at various points, or if the authors are referring to high numbers (or %) of CD169+ SMs in lymph nodes. 

Author Response

We sincerely appreciate for kind review and helpful comments. We added the sentences as reviewer’s comments/advises. 

Comment 1: First sentence: Please revise, as there are more types of cancer antigens than those that derive from neoantigens and cancer testis antigens.

>> We changed the pointed sentences as follows.

Cancer cells are characterized by the accumulation of a variable number of genetic alterations that result in the production of neoantigens. Cancer specific antigens such as cancer-testis antigens, oncofetal antigens, aberrantly expressed proteins, and viral antigen also are target of cytotoxic T cells.

In section 6, it is a little unclear if the authors are referring to high CD169 expression on SMs at various points, or if the authors are referring to high numbers (or %) of CD169+ SMs in lymph nodes.

>> As reviewer pointed, readers might confuse the two kinds of words “high expression of CD169” and “high number of CD169-positive macrophages”. We used only “high number of //” in the text as follow.

“Statistical analyses of postoperative survival and clinicopathological factors between patients with high and low number of CD169-positive SMs in lymph nodes were also performed.”

Reviewer 3 Report

This is a short review of the contributions of lymphatic macrophages (specifically CD169+ cells) to tumor development. The authors briefly review the contributions others have made on the subject (including their own work) and provide some new data to support this. The review is short and may not include all of the recent citations on this subject, the reviewer has outlined a couple that may be of interest below. Overall, the review was well-written and flowed well.

Comments below:

Page1L1: Cancer testis antigens – do you mean cancer antigens more generally?

Page2: “Targeting antigen delivery to CD169-expressing cells was shown to enhance immune responses in pigs” – Are there any studies showing this in humans or mice? Please cite here as well.

This study focuses on CD169+ macrophages as the marker for lymphatic macrophages so the abstract should reflect this.

Fig 1: Why wasn’t CD169 expression tested as a proof of principle?

A table with the data for variable expression of CD169 in human lymph nodes would be useful.

Fig. 2 - the % cover in should be reported for the immunohistochemical analysis. Were multiple samples taken and if so, what were their means/stdv? Also, the scalebars says this is in meters not microns? This may have been a pdf conversion error.

Fig 3. – It looks like PD-L1(+) SMs and DCs are inhibiting CD8+ T-cells in this figure. I might also include the effect CD8+ T-cells have on the tumor cells in this figure as well.

Grabowska et al. are doing a lot of work in this area, you may want to view their work.

For example:  Grabowska J, Lopez-Venegas MA, Affandi AJ, den Haan JMM. CD169+ Macrophages Capture and Dendritic Cells Instruct: The Interplay of the Gatekeeper and the General of the Immune System. Front Immunol. 2018;9:2472. Published 2018 Oct 26. doi:10.3389/fimmu.2018.02472

 Also you may be interested in this recent review:

Liu, Yu, Yuan Xia, and Chun-Hong Qiu. "Functions of CD169 positive macrophages in human diseases." Biomedical Reports 14.2 (2021): 1-1.

Author Response

We sincerely appreciate for kind review and helpful comments. We added the sentences as reviewer’s comments/advises. 

Comment 1; Page1L1: Cancer testis antigens – do you mean cancer antigens more generally?

>> We corrected this part as follow;

“Cancer cells are characterized by the accumulation of a variable number of genetic alterations that result in the production of neoantigens. Cancer specific antigens such as cancer-testis antigens, oncofetal antigens, aberrantly expressed proteins, and viral antigen are also target of cytotoxic T cells.”

Comment 2; Page2: “Targeting antigen delivery to CD169-expressing cells was shown to enhance immune responses in pigs” – Are there any studies showing this in humans or mice? Please cite here as well.

>> Few studies other than mice have been published, we cited the study of pig. Many studies using mice were published and already cited in the text.

Comment 3; This study focuses on CD169+ macrophages as the marker for lymphatic macrophages so the abstract should reflect this.

>> We added some words in the abstract.

Comment 4; Fig 1: Why wasn’t CD169 expression tested as a proof of principle?

>> Picture figure of CD169 expression in murine lymph node was added in the figure 1.

Comment 5; A table with the data for variable expression of CD169 in human lymph nodes would be useful.

>> We summarized this point in the Table 1.

Comment 6; Fig 3. – It looks like PD-L1(+) SMs and DCs are inhibiting CD8+ T-cells in this figure. I might also include the effect CD8+ T-cells have on the tumor cells in this figure as well.

>> As reviewer commented, Figure 3 was not enough completed. We added some points in the figure 3.

Grabowska et al. are doing a lot of work in this area, you may want to view their work.

For example:  Grabowska J, Lopez-Venegas MA, Affandi AJ, den Haan JMM. CD169+ Macrophages Capture and Dendritic Cells Instruct: The Interplay of the Gatekeeper and the General of the Immune System. Front Immunol. 2018;9:2472. Published 2018 Oct 26. doi:10.3389/fimmu.2018.02472 Also you may be interested in this recent review: Liu, Yu, Yuan Xia, and Chun-Hong Qiu. "Functions of CD169 positive macrophages in human diseases." Biomedical Reports 14.2 (2021): 1-1.

>> We added the following sentence as cited of Front Immunol. 2018;9:2472. Biomedical Reports 14.2 (2021) has already cited in the text.

“Antigens circulating in the blood flow preferentially captured by spleen antigen presenting cells including CD169+ macrophages which transfer the antigens to B cells and DCs [21,22].”

Round 2

Reviewer 1 Report

The authors made changes per my comments. I suggest acceptance.